# Critical Perspectives on the Design of Polymeric Materials for Mitigating Thermal Runaway in Lithium-Ion Batteries

**DOI:** 10.3390/polym17091227

**Published:** 2025-04-30

**Authors:** Hangyu Zhou, Jianhong He, Shang Gao, Xuan Cao, Chenghui Li, Qing Zhang, Jialiang Gao, Yongzheng Yao, Chuanwei Zhai, Zhongchun Hu, Hongqing Zhu, Rongxue Kang

**Affiliations:** 1China Academy of Safety Science and Technology, Beijing 100012, China; gao921911@163.com (S.G.); cx53535@126.com (X.C.); qzhang@163.com (Q.Z.); yaoyz@cumtb.edu.cn (Y.Y.); hzc.32@163.com (Z.H.); 2National Academy of Safety Science and Engineering, Ministry of Emergency Management of the People’s Republic of China, Beijing 100012, China; chli@163.com (C.L.); jlgao@163.com (J.G.); 18266085850@163.com (C.Z.); 3Key Laboratory of Electrochemical Energy Safety, Ministry of Emergency Management of the People’s Republic of China, Beijing 100012, China; 4School of Emergency Management and Safety Engineering, China University of Mining and Technology, Beijing 100083, China; hejianhong454@163.com (J.H.); zhq@cumtb.edu.cn (H.Z.); 5National Institute of Natural Hazards, Ministry of Emergency Management of the People’s Republic of China, Beijing 100085, China

**Keywords:** lithium-ion batteries, thermal runaway, polymeric materials, multi-parameter sensing, safety mechanisms

## Abstract

During the global energy transition, electric vehicles and electrochemical energy storage systems are rapidly gaining popularity, leading to a strong demand for lithium battery technology with high energy density and long lifespan. This technological advancement, however, hinges critically on resolving safety challenges posed by intrinsically reactive components particularly flammable polymeric separators, organic electrolyte systems, and high-capacity electrodes, which collectively elevate risks of thermal runaway (TR) under operational conditions. The strategic integration of smart polymeric materials that enable early detection of TR precursors (e.g., gas evolution, thermal spikes, voltage anomalies) and autonomously interrupt TR propagation chains has emerged as a vital paradigm for next-generation battery safety engineering. This paper begins with the development characteristics of thermal runaway in lithium batteries and analyzes recent breakthroughs in polymer-centric component design, multi-parameter sensing polymers, and TR propagation barriers. The discussion extends to intelligent material systems for emerging battery chemistries (e.g., solid-state, lithium-metal) and extreme operational environments, proposing design frameworks that leverage polymer multifunctionality for hierarchical safety mechanisms. These insights establish foundational principles for developing polymer-integrated lithium batteries that harmonize high energy density with intrinsic safety, addressing critical needs in sustainable energy infrastructure.

## 1. Introduction

Lithium-ion batteries (LIBs) possess excellent energy density and outstanding cycle life, making them a core technology for achieving “dual carbon” goals and facilitating the energy transition [1,2]. However, lithium batteries will incur thermal runaway (TR) under the influence of both internal and external factors. LIBs contain integrated flammable polymers, organic electrolytes, and high energy density electrodes, which can easily lead to fires and explosions without targeted preventive measures, posing a threat to the safety of people’s lives and property (Figure 1a) [3,4]. Once thermal runaway is initiated, external measures often struggle to address the internal core exothermic reactions, and the violent combustion chain reaction of polymers leads to inefficient risk management.

As evidenced by 23 documented global incidents (2023–2024 Q1, Table 1), delayed TR intervention results in irreversible consequences: 78% of cases exhibited <3-min containment windows post-ignition. Current mitigation strategies face fundamental limitations—external thermal management cannot suppress self-accelerating internal reactions, while component decomposition and polymer combustion generate sustained thermal feedback (Q > 2.5 kW/cell).

The internal causes of TR mainly refer to issues arising during the design and manufacturing process of the battery, with internal short-circuits being a primary factor. Mechanical abuse, electrical abuse, and thermal abuse are the three main categories of external factors that trigger thermal runaway in batteries and are the leading causes of this phenomenon [5]. Therefore, the development of efficient and stable TR suppression technologies is a crucial research direction for enhancing the thermal safety of LIBs (Figure 1b) [6,7]. In general, the TR process is associated with changes in the characteristic parameters of LIBs, such as temperature, stress, voltage, resistance, and gas emission [8,9]. Developing responsive materials with high sensitivity and excellent switch ratios is a crucial strategy for enhancing the safety of lithium batteries. As research deepens into the temporal relationships of TR behavior and characteristic parameters in lithium batteries, multi-parameter responsive intelligent safety materials have expanded into domains such as battery thermal management, electrode materials, separators, and electrolytes [10,11]. Capitalizing on their tailorable molecular architectures, polymeric materials demonstrate exceptional versatility in multi-stimuli responsive applications. Furthermore, each constituent component within lithium-ion batteries (LIBs) necessitates a unique set of design criteria that must be meticulously addressed through systematic material engineering to achieve optimized performance metrics. This review recalls the research progress on the characteristics of TR in lithium batteries, the types of intelligent safety materials, and the mechanisms of multi-parameter response. It analyzes the challenges faced by intelligent safety materials in the prevention and control of TR in lithium batteries and looks ahead to research directions for material design and technological applications in new battery systems and complex scenarios. The findings aim to provide scientific guidance for promoting the safe design of lithium batteries.

## 2. Thermal Runaway Characteristics

TR in lithium batteries is a major safety issue faced by electrochemical energy storage systems. TR refers to incidents where electrochemical batteries uncontrollably increase their temperature through self-heating [12,13]. Multiple standards contain definitions regarding thermal runaway and its propagation (Table 2). Lithium batteries are highly integrated and high-energy units; once TR occurs, external measures struggle to reach the internal core reactions, while the combustion chain reaction from polymers, liquid electrolytes, and other batteries‘ components makes it difficult to effectively contain the spread of the disaster. Therefore, studying the evolution mechanism of battery thermal runaway is of utmost importance.

### 2.1. Thermal Runaway of LIBs

Generally, thermal runaway can be divided into three stages: the self-heating stage (50–140 °C), the thermal runaway stage (140–850 °C), and thermal runaway termination (850 °C to ambient temperature) (Figure 2a) [18,19,20]. The self-heating stage is triggered by solid electrolyte interphase (SEI) decomposition at ≈90 °C due to the exothermic reactions between lithiated anode and electrolyte solvents. The metastable SEI dissolution exposes fresh anode surfaces, accelerating redox reactions that establish a positive feedback loop for heat accumulation. Once the temperature exceeds 140 °C, both the positive and negative electrode materials participate in electrochemical reactions, causing the battery temperature to rise rapidly. Critical phase transitions occur as separator meltdown (≈130–160 °C) induces large-area internal short circuits, accompanied by voltage collapse. Concurrently, cathode decomposition releases lattice oxygen (LiCoO_2_ → CoO + ½O_2_), while electrolyte decomposition generates combustible gases (CO, CH_4_, HF). These exothermic chain reactions drive temperature escalation rates exceeding 10 °C/s, resulting in the propagation of thermal runaway. The system reaches peak reaction intensity with violent venting of cell contents, followed by gradual cooling as reactant depletion terminates exothermic processes. Once thermal runaway occurs, the process can only naturally terminate once the reactants are exhausted.

Notably, the thermodynamic sequence exhibits characteristic electrical signatures (Figure 2b): progressive SEI degradation manifests as subtle voltage fluctuations, while separator failure induces abrupt voltage drop (>130 °C). Real-time monitoring of coupled electrical-thermal parameters (dV/dT, impedance phase shift) enables predictive TR identification 30–60 s before catastrophic failure. This critical detection window permits activation of multilayer safety protocols, including polymer-based current interrupt devices and fire-suppression electrolytes to mitigate propagation risks in battery packs.

### 2.2. Triggering Mechanism of Thermal Runaway in LIBs

Thermal runaway is caused by two categories of internal factors and external factors. Internal factors primarily refer to the issues arising during the design and manufacturing processes of the battery, while external factors pertain to causes related to personnel and external conditions during transportation, installation, and operational maintenance of the battery. Internal short circuits are a major internal cause of thermal runaway. Due to the varying degrees of contact between the positive and negative electrodes of the battery, the subsequent reactions triggered can differ significantly. As shown in Figure 3, internal short circuits induced by defects generated during battery design and manufacturing have relatively mild heat generation in the initial stage, leading to no significant voltage drop. However, as the battery ages and various performance degradations occur (such as increased internal resistance and lithium metal deposition), the risk of internal short circuits gradually increases, leading to the accumulation of heat and noticeable voltage decline. If effective prevention and control measures are not in place, the battery will trigger thermal runaway.

Mechanical abuse, electrical abuse, and thermal abuse are the three main external factors that trigger thermal runaway in batteries. Mechanical abuse refers to the deformation of battery cells or packs under external forces, with abusive forms including impacts, compression, and puncturing in safety evaluation tests [21,22]. Electrical abuse typically includes forms such as external short circuits, overcharging, and over-discharging, with overcharging being particularly prone to developing into thermal runaway. Under overcharging conditions, the generation of heat and gas are two significant characteristics. Due to excessive lithium insertion, lithium dendrites grow on the surface of the anode. The stoichiometric ratio between the cathode and anode determines the growth sequence characteristics of the lithium dendrites. Excessive de-intercalation of lithium causes the cathode structure to collapse due to heat generation and oxygen release, while the release of oxygen accelerates electrolyte decomposition, generating a large amount of gas and increasing internal pressure, leading to the opening of the vent valve for gas release [23]. Once the active materials in the battery come into contact with air, a vigorous reaction occurs, releasing a substantial amount of heat. Therefore, overcharging is the most hazardous form of electrical abuse. Thermal abuse rarely occurs independently; it often develops from mechanical and electrical abuse. Local overheating is a typical situation that can occur within a battery pack, and it is a critical link that can directly trigger TR. In addition to overheating caused by mechanical or electrical abuse, thermal abuse caused by loose connections has also been confirmed by relevant studies [24]. Thermal abuse is currently the most frequently simulated failure condition in batteries, and the use of adiabatic calorimeters is an important method for researching the TR process and triggering mechanisms in batteries [25].

**Figure 3 polymers-17-01227-f003:**
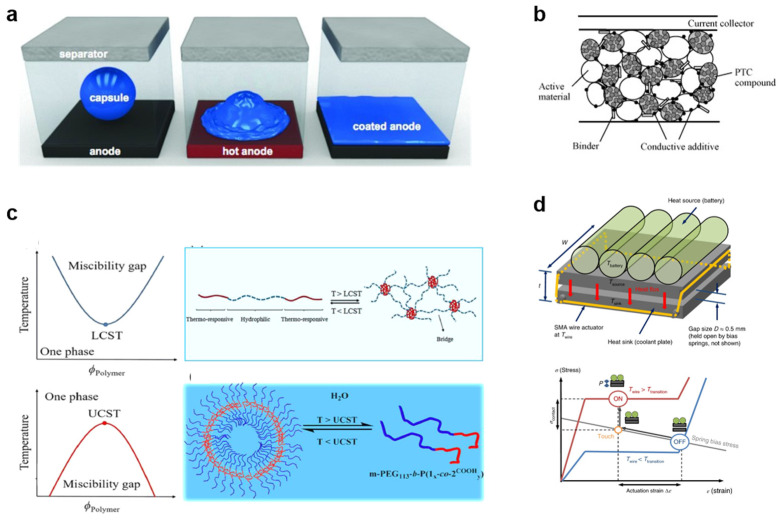
Categories of smart safety materials: (**a**) phase change materials [26] (Copyright 2020 Elsevier); (**b**) positive temperature coefficient materials [27] (Copyright 2007 Elsevier); (**c**) sol-gel transition polymers [28] (Copyright 2019 Elsevier); (**d**) shape memory materials [29] (Copyright 2018 Nature).

## 3. Categories of Smart Safety Materials

Safety is a critical issue facing lithium batteries, and traditional thermal protection measures primarily rely on temperature parameter responses, which pose challenges to efficiency of shutdown. Exploring smart safety materials that respond to characteristics such as temperature, light, force, and electricity to enhance the disaster perception and response capabilities of lithium batteries is an important research direction for achieving high safety in lithium batteries. This review introduces the applications of phase change materials (PCMs), positive temperature coefficient materials (PTCs), shape memory materials (SMMs), and sol-gel transition polymers (STPs) as smart safety materials for lithium batteries.

### 3.1. Phase Change Materials

PCM is a class of typical functional material that includes polymers and hydrated salts, capable of storing energy by absorbing heat during melting and releasing heat during solidification [26,28]. It has a significant latent heat and a small temperature range for phase change, maintaining the phase change process under isothermal or near-isothermal conditions. Benefiting from this characteristic of PCM, it is possible to manage the internal heat of the battery, ensuring a stable and moderate operating environment, suppressing the propagation of thermal runaway, and enhancing safety.

### 3.2. Positive Temperature Coefficient Materials

PTC exhibits an increasing resistance during rising temperature and rarely deteriorate properties at normal temperatures in operating batteries. Polymers are typical PTC that can response to temperature fluctuation. The degree of resistance increase under a transition temperature is a crucial requirement for protection against overheating in LIBs. Recent studies have proposed some PTC components with a proper transition temperature, considerable conductivity, and obvious shutting down effect, as well as excellent electrochemical and chemical stability in operating batteries, enabling the efficient prevention of thermal runaway [30,31]. Therefore, PTC composites can be utilized as electrodes, electrolytes, current collectors, and separators. These properties of PTC materials provide an approach for building much safer batteries.

### 3.3. Sol-Gel Transition Polymers

STP is a class of polymers capable of undergoing phase separation at a critical temperature [27,29]. Common synthesis methods include reversible addition-fragmentation chain transfer polymerization (RAFT), atom transfer radical polymerization (ATRP), and free radical polymerization (FRP). Based on whether they exhibit a lower critical solution temperature (LCST) or an upper critical solution temperature (UCST), sol-gel transition polymers can be categorized into two types. LCST refers to the lowest temperature on the binary curve of the phase diagram, above which two phases exist. In contrast, UCST indicates the temperature at which the bimodal curve reaches its maximum; below this temperature, two phases are present. By utilizing the aforementioned characteristics of STP, it is possible to modulate the transition temperature (LCST and UCST) of the polymers through molecular design, achieving reversible blocking and recovery of the mass transfer process in lithium battery electrochemical reactions, thereby ensuring the safe and stable operation of the battery.

### 3.4. Shape Memory Materials

SMM is a class of functional polymer materials that combine properties of both plastics and rubber [32,33]. Through molecular design and modification, these polymer materials are endowed with desired deformation characteristics. When external conditions change, they can correspondingly alter their shape and fix that altered (deformed) state. If the external environment changes again in a specific manner and pattern, they can reversibly return to their original state. By utilizing the “memory of the initial state-fixed deformed state-recovery to the initial state” characteristic, lithium battery key components designed and constructed in this manner can respond to characteristic signals such as temperature, pressure, and current, enhancing the disaster perception and handling efficiency of lithium batteries and effectively suppressing the spread of disasters.

In general, smart safety materials suitable for lithium batteries need to meet the following requirements: (1) they should have appropriate response windows for thermal, mechanical, and electrical parameters, along with excellent switching ratios; (2) they must possess good thermal stability to endure harsh operating temperatures; (3) they should exhibit outstanding chemical and electrochemical stability to broaden the design and application of key components in lithium batteries.

## 4. Design and Application of LIBs’ Key Components

The application of smart safety materials is an important measure for enhancing the safety of lithium batteries. This paper reviews the significant roles played by thermal-responsive, electric-responsive, stress-responsive, and other safety materials in improving the safety and electrochemical performance of lithium batteries from the perspectives of design paradigms, component structures, and functional types.

### 4.1. Thermo-Responsive Safety Materials

Thermal signal emerges during the thermal runaway process and reflects the hazard status in operating batteries. Therefore, thermo-responsive materials are crucial safety components, forming an important research aspect for enhancing batteries’ safety and stability [34].

#### 4.1.1. Thermo-Responsive Electrodes

In general, high energy density inherently presents elevated safety risks; reported electrode materials with high energy densities, such as lithium transition metal composite oxides (LiCoO_2_, LiNi_0.5_Mn_1.5_O_4_, LiNi_0.8_Mn_0.1_Co_0.1_O_2_ and so on), sulfur, and lithium metal still suffer from a limited lifespan due to continuous reconstitution of the electrode–electrolyte interface, resulting in spontaneous heat accumulation [35,36]. Electrical and ionic blocking are typical designs for internal application of thermo-responsive materials, achieving a crucial factor for shutting down electrochemical reactions under internal/external overheat [37]. PTC exhibit a distinct resistance increase undergoing elevated temperature and shut down the electrochemical reaction through an electrical blocking effect. However, constructing a highly conductive matrix in electrode materials (up to 4 × 10^4^ Ω cm^−1^) is still difficult, because that PTC are generally synthesized with non-conductive monomers. In order to address this challenge, PTC designed with enhanced intrinsic conductivity are proposed. Li et al. proposed a designed PTC polymer by mixing poly(3-octylpyrrole):poly (styrenesulfonate) (P3OPy:PSS) and carbon composite (P3OPy:PSS/C) [38]. P3OPy:PSS achieves a high conductivity as it is PSS anionic-doped and promotes charge transfer in operating batteries. Once the internal temperature exceeds the normal operating window, (P3OPy:PSS)/C can efficiently transfer to insulation and further block the charge transference at the cathode–conductive agent interface (Figure 4a). Fabricated in this way, the graphite||LiCoO_2_ pouch cell has demonstrated considerable electrochemical performance and avoided TR even at 120 °C. Zhang et al. developed an integrative cathode with PTC effect by constructing a redox-active P3BT layer between the LiNi_0.5_Co_0.2_Mn_0.3_O_2_ (NCM523) coating and the aluminum foil current collector (Figure 4b) [39]. This integrative cathode delivers an excellent charge/discharge capacity of 141.7 mAh g^−1^ at operating temperature, achieving a rapid voltage drop by sharply increasing resistance at 120–150 °C. As expected, the electrochemical reaction in the pouch cell is completely terminated under persistent overheating conditions (150 °C) for 5 min. In order to reduce side reactions of PTC materials and organic electrolytes and improve safety redundancy, lower thermo-responsive triggered temperature is urgently needed. Therefore, a highly fluoridated system has been proposed. Jin et al. constructed a poly(vinylidene fluoride) (PVDF)-based PTC structure with carbon-coated LiFePO_4_ and super-P (Figure 4c) [40]. This conductive cathode network exhibits rapid electron transfer at operating temperature and achieves a PTC effect due to the blocked electronic pathway caused by the sharp volume expansion of PVDF at 80 °C. Polyethylene (PE) exhibits a melting point of 105 °C, with recent studies demonstrating that the sandwich-like separator of PP/PE/PP can produce an ionic blocking effect by closing the ion transport channels in PP [41]. Baginska et al. prepared a thermo-responsive coating on a graphite anode substrate, where the as-synthesized PE microspheres, with an average diameter of 4 μm, guarantee fluent ion transport at the graphite–electrolyte interface under normal working temperatures. However, their thermal transformation, specifically melting, creates ionic insulation barriers that shut down the electrochemical reaction at a critical temperature [42]. In order to achieve smart modification of batteries, reversibly thermo-responsive materials have been widely explored and highly integrated. Chen et al. reported a reversibly thermo-responsive cathode made with a built-in low-density polyethylene (LDPE) that has a thermal expansion coefficient of around 10^−4^ K^−1^ [43]. As seen in Figure 4d, a reversibly thermo-responsive layer was assembled by mixing molten LDPE and graphene-coated spiky Ni nanoparticle filler, which guarantees rapid electrical transport between the cathode and the current collector. Benefiting from reversible expansion and contraction, this smart layer achieves a maximal reduction in conductivity by 7–8 orders of magnitude during thermal abuse and increases to a high conductivity of 50 S cm^−1^ when the internal temperature recovers to suitable operating conditions, providing a smart modification effect for safer batteries undergoing overheating.

#### 4.1.2. Thermo-Responsive Electrolytes

Flammable liquid electrolytes constitute a significant safety concern of TR, as they can easily be ignited under accumulated heat and oxygen conditions, leading to fires or explosions [44]. Therefore, electrolytes designed with reversibly thermo-responsive features need to be applied to improve battery safety. Sol-gel transition polymers with a suitable critical temperature can restrain early TR risk. Kelly et al. synthesized a PNIPAM-AA copolymer electrolyte through free radical polymerization with an initiator, which can control thermal performance and provide electrolyte ions (Figure 5a) [45]. In full-cell testing, the overall cell capacity was reduced by about 85% during the temperature increase from 20 °C to 50 °C due to its unique LCST, which prevents internal overheating. When the temperature cools down, the polymer dissolves into the solution again, releasing free ions for a continuous electrochemical reaction. Conventional electrolytes that use the sol-gel transition strategy still face numerous problems, such as packing difficulties, phase separation, and liquid leakage. Therefore, gel polymer electrolytes with reversibly thermo-responsive features have been developed. Zhang et al. selected a free radical copolymerization strategy to obtain the poly(N-isopropylacrylamide-co-N-methylolacrylamide) (PNIPAM/NMAM) hydrogel polyelectrolyte [46]. The full cell using this electrolyte delivers considerable electrochemical properties at room temperature and exhibits a maximum capacity loss rate of more than 80% at 70 °C, reflecting the dynamically thermally self-protected feature for control and prevention of TR. Temperature-stimulus SMPs display the capability to temporarily fix a shape under specific external thermal activation conditions and revert to their original shape after cooling down. Ureidopyrimidinone (UPy)-containing polymers are typical units for constructing shape memory polymer electrolytes. Jo et al. fabricated an SPE with poly(vinyl alcohol) (PVA) as the main chain and epoxy-functionalized poly(ethylene glycol) (PEG) and UPy as the side chains [47]. Benefiting from the physical cross-linking network formed via quadruple hydrogen bonding, the PVA-UPy-PEG polymer electrolyte can transform from a temporary shape to a permanent shape under a heat stimulus (60 °C). The battery using this electrolyte shows not only efficient overheat protection but also outstanding electrochemical performance. However, SMPs still face complicated synthesis routes, inferior tensile stress (~40 kPa), and inefficient separation processes. Thus exploring SMP categories with shape recovery and tunability is a crucial topic for developing next-generation safe and high-performance energy storage technologies.

In practical applications, LIBs usually face extreme safety hazards, such as uncontrollable heat accumulation. Therefore, the thermal trigger temperature and shutdown efficiency are crucial parameters. As seen in Figure 5b, Zhou et al. designed a smart thermo-responsive electrolyte (PPE) through poly(ethylene glycol) methyl ether methacrylate (PEGA) and 2,2,3,3,3−pentafluoropropyl acrylate (PFE) copolymerization [48]. This smart thermo-triggered electrolyte exhibits an excellent thermal stability that elevates batteries’ operating temperature up to 100 °C and achieves rapid transformation from liquid to solid under 130 °C, which is caused by its thermal free radical polymerization behavior without any short-circuit or gas expansion. Even at temperatures of up to 150 °C, LIBs integrated with PPE do not suffer from short-circuit failure. In addition, LIBs integrated with PPE exhibit an excellent thermal reversibility that maintains a capacity retention of 95% after 200 cycles at 0.1 °C after cooling down (30 °C).

#### 4.1.3. Thermo-Responsive Current Collectors

A current collector is a bridge for charge transfer in LIBs. Jia et al. designed an innovative current collector built with shape memory polymers for TR inhibition [49]. The configuration of the current collector is shown in Figure 6a, with a highly conductive Cu layer deposited onto the shape memory polymer surface via magnetron sputtering. This complex current collector exerts no influence on electrochemical properties during regular battery operation, while rapidly changing its original shape when undergoing internal TR (>90 °C), leading to a transformation from a conductive to an insulating state in a sacrificial manner. As depicted in Figure 6b, the full cell provides stable operating conditions at moderate temperatures (30–50 °C) and exhibits swift self-closure under overheating conditions at 120 °C. This strategy renders the post-heating event completely inoperable, demonstrating its effectiveness and ensuring safety, especially in a high-nickel ternary system. Though electrical and ionic blocking are typical designs for inhibiting TR, combustion and explosion risks persist due to the absence of internal fire-extinguishing means. To build safer batteries, traditional approaches prefer to embed flame retardant materials in the current collectors. However, this method tends to, sacrifice the transport of lithium ions or electronic conduction capability to some extent. To bolster the overall energy density of LIBs through current collector design, Ye et al. fabricated an ultralight thermo-responsive current collector with a porous polyimide (PI) substrate [50]. As seen in Figure 6c, porous PI is filled by triphenyl phosphate (TPP) and coated with copper. With the low melting point of TPP (around 50 °C), the response time to TR can be significantly shortened. In terms of safety, the self-extinguishing time (SET) value significantly decreases as the thickness of PI-TPP-Cu increased from 5 to 9 μm. As for the whole electrode, the PI-TPP-Cu-based graphite electrode achieved self-extinguishing completely within 1.0 s. These findings suggest that the as-fabricated PI-TPP-Cu sandwich current collector exhibits remarkable and lightweight thermal stability (>400 °C) and notable flame retardancy.

#### 4.1.4. Thermo-Responsive Separators

With growing demand for high energy density in cutting-edge battery technologies, multi-functional separators have attracted significant interest in recent years [51,52]. In order to promote the shutdown effect of LIBs, recent studies have focused on appropriate critical temperature, rapid response speed at critical temperatures, and excellent chemical stability within the battery system.

Commercial Celgard2325 (PP/PE/PP) membrane is a typical thermo-responsive separator where the PE layer can impede ion transport by closing hole above 115 °C and inhibit TR propagation [53]. However, such a high thermo-responsive temperature will directly increase the risk of TR. Constructing a separator that features a lower melting point is a feasible approach to substituting the commercial PP/PE/PP membrane. In general, thermal-responsive polymers can also be introduced into LIBs for functional separators. Dong et al. synthesized a functional coating of core–shell paraffin@SiO_2_ particles [54]. Applied to PE, this coating exhibits favorable ion diffusion and compatibility with commercial electrolytes and releases core paraffin under TR, ensuring the stable and safe electrochemical performance of working batteries. Ji et al. synthesized a new thermal shutdown separator by coating thermoplastic ethylene–vinyl acetate copolymer (EVA) microspheres onto PE surfaces [55]. The thermal-responsive EVA micropores can melt and blocked channels of PE at 90 °C, achieving Li^+^ blocking in working batteries. These processes efficiently expand the application conditions of polyolefins for high-energy-density battery systems. Not restricted to polyolefin separators, a three-layer non-woven separator featuring an amide-functionalized polyetheretherketone (APEEK) outer layer also exhibits a broad shutdown temperature range from 100 to 270 °C, effectively ensuring the battery’s safety [56]. Shen et al. built a thermal-responsive separator containing a poly(sulfobetaine) polymer (PMABS) with a high upper critical solution temperature (UCST) and graphene oxide (GO) sheets with electronic insulation [57]. The thermal cutoff in the battery at regular operating temperature is displayed in Figure 7a with the dipole attraction between zwitterions resulting in the tight coiling of the polymer and forming hydrophobic polymer attachments on the membrane, enabling open channels for rapid ions transferring. Beyond the UCST, the disruption of dipole attraction between zwitterions leads to the unfolding of polymer chains and exposes the zwitterions to the electrolyte. This results in an elevated electrolyte viscosity within the membrane, which achieves thermal cutoff in the battery by limiting the migration of lithium ions. However, phase conversion tactics (sol–gel) still face leakage of polymers during the re-cooling process, resulting in increased interface impedance and sacrificial properties in batteries. Electrospinning technology is a convenient way to fabricate porous polymer skeletons with varying surface areas and pore structures [58]. At present, polytic fluoride-hexarinol (PVDF-HFP), polyimide (PI), and other polymers have been widely used as substrates [59,60]. Jiang et al. have proposed a polylactic acid (PLA) @ polybuthylenesuccinate (PBS) core–shell separator made using an electrospinning technology [61]. PLA core delivers excellent thermal stability and mechanical strength, and the PBS shell exhibits a strong electrolyte affinity and proper melting temperature (130 °C). LIBs with this separator exhibit outstanding electrochemical properties at regular operating temperatures and a self-protective effect under overheating conditions.

Though numerous reports have focused on switching functions for smart partitions under high-temperature conditions, their reversibility remains a challenge for reusable batteries. Jiang et al. synthesized a temperature-dependent PVP@TiO_2_ separator using a modified electrospinning method, achieving excellent wettability toward liquid electrolytes and maintaining structural integrity [64]. Once the battery’s internal temperature exceeds 60 °C, PVP reacts with the liquid electrolyte and blocks the pores in the separator, thus terminating the electrochemical reaction. When the temperature returns to normal operating conditions, the battery’s capacity is restored because the TiO_2_ skeleton restores the structure. The inorganic Al_2_O_3_, SiO_2_, ZrO_2_, and zeolite particles also exhibit potential reversible responses when polyvinyl alcohol (PVA) is used as a binder [65]. In addition, the as-fabricated functional separator shows enhanced thermal stability compared to commercial PP separators. For inorganic-organic systems, it is imperative to carefully select an appropriate organic polymer as a binder to ensure the stability of the composite coating.

In order to deal with potential combustion and explosion risks of high-energy-density batteries, positioning flame retardant additives within the separator has become an effective approach. For example, Yim et al. designed a core-shell structural coating with 1,1,1,2,2,3,4,5,5,5−decafluoro-3-methoxy-4-(trifluoromethyl)-pentane (DMTP) as a fire-extinguishing agent (Figure 7b) [62]. Under mechanical abuse (nail-puncture test), a 0.5 Ah pouch cell with a commercial PE separator exhibits a rapid temperature surge (72.3 °C), whereas with a PE coated with a DMTP-containing coating shows a markedly reduced maximum temperature of 37.2 °C, demonstrating a significant improvement in safety. The core–shell structure can avoid parasitic reactions between the flame retardant additive and the active electrode, releasing the flame retardant additive at critical temperatures. As displayed in Figure 7c, Liu et al. encapsulated triphenyl phosphate (TPP) into a polyvinylidene-hexafluoropropylene (PVDF-HFP) shell using electrospinning technology [63]. This microfiber separator rarely affects battery properties under mild operating conditions. When TR spreads violently, the PVDF-HFP shell melts at 160 °C and releases TPP into the electrolyte, effectively inhibiting electrolyte combustion.

Through thermo-responsive components that contain polymeric materials, the tisk of TR under overheating conditions within LIBs can alleviate. However, these strategies are not ideal for practical application scenarios because they cannot entirely eliminate the onset of TR with further heat generation from electrode reactions. Hence, to achieve higher levels of thermal safety, strategies that block ion/electron transport to stop the operation of LIBs upon early anomalous signals have also been developed to shut down heat generation and prevent possible TR.

### 4.2. Electric-Responsive Safety Materials

For LIBs, external short-circuit (EST), overcharging, and over-discharging are typical electrical abuse behaviors. Among these, overcharging poses the highest risk of transitioning into TR [66,67]. Therefore, recent studies have generally focused on overcharge protection in electric-responsive safety materials design. Components such as cathode coatings, electric-responsive electrolytes, and potential response separators have been developed to enhance the anti-overcharge behavior of LIBs [68].

#### 4.2.1. Electric-Responsive Additives

Electro-polymerization additives exert a passivation effect at the cathode and disrupt internal ion transport, preventing the continuous decomposition of the electrolyte and the onset of TR in the batteries [69]. Biphenyl (BP) is one of the earliest polymer additives for overcharge protection. When overcharging occurs in the battery, BP can generate a sleek and dense polymer film through electro-polymerization on the electrode surface. Xiao et al. successfully addressed overcharging by introducing a 1 M LiPF_6_ EC/DMC (1/1) electrolyte with 2.5 wt.% BP [70]. Xu et al. proposed cyclohexyl benzene (CHB) as an electro-polymerization additive, although it only provides reliable overcharge protection for the battery under reduced operating voltages [71]. Korepp et al. have determined a series of additives with high electrochemical polymerization potentials (benzyl isocyanate (BIC) for 5 V and 4-bromo-benzyl isocyanate (Br-BIC) for 5.5 V), which are suitable for high-voltage cathode systems [72]. The experiments suggest that the electrochemical polymerization additive can effectively suppress TR even at a low concentration (2 wt.%) in a 1 M LiPF6 EC/DMC (1/1) electrolyte. This result reflects that regulating functional groups on the benzene ring of polymerized anti-overcharge additives is a strategy to enhance their performance, broadening their sequence such as diphenylamine (DPAn), Nphenylmaleimide (NPM), dimethoxydiphenylsilane (DDS), bis(diphenyl phosphate) (RDP), and (2-chloro-4-methoxy)-phenoxy pentafluorocyclotriphosphazene (2-Cl-4-MPPFPP) [73,74,75,76,77]. In general, electro-polymerization protection during overcharge is irreversible. Therefore, there is an increased need for developing electro-polymerization protection systems with high reversibility.

Redox shuttle additives are typical electro-polymerization materials with reversibility. The operating mechanism of redox shuttle additives involves two steps: their conversion into an oxidized form [O] on the overcharged anodic electrode; and their restoration to their original state [R] on the cathodic electrode, which maintains the ‘‘oxidation-diffusion-reduction-diffusion” loop driven by diffusion [78]. This process can effectively constrain the anodic electrode potential at the oxidation state until the excess charge has been completely consumed with no damage to the battery’s capacity. Behl et al. first revealed the redox shuttle LiI-I_2_ loop in a 1.5 M LiAsF_6_ tetrahydrofuran (THF) electrolyte [79]. LiI can be oxidized to I_2_ under overcharging conditions and effectively prevents the oxidation of solvent molecules. Subsequently, I_2_ can diffuse to the cathodic electrode and reduce to reduction products of LiI. Benefiting from this loop in the electrolyte, the full cell can steadily operate without the distress of TR caused by overcharging. At present, numerous redox shuttle additives have been clarified, including ferrocene, polypyridine, dimethoxybenzene, phenothiazine, and their derivatives and mechanisms [80,81,82,83].

Most metallocene compounds are easily miscible with organic electrolytes, ensuring good stability, easy preparation, and low costs. Abraham et al. introduced various metallocene compounds with diverse metal centers into the electrolyte of 1.5 M LiAsF6 in THF: 2MeTHF: 2MeF (volume ratio is 48:48:4), and found that regulating the metallic atoms or substituents on the cyclopentadienyl rings can alter the overcharge prevention performance [84]. Golovin et al. also suggested that tailoring various substituent groups of ferrocene derivatives exerts a significant influence on the electrochemical performance, diffusion constants, and shuttle voltage range of the full cell [85]. These results show that electron-withdrawing substituents entail a higher oxidation potential, whereas electron-donating substituents result in a lower oxidation potential, which is highly consistent with theoretical predictions, as a higher electron cloud density results in a higher HOMO in the molecule. Although metallocene and its derivatives exhibit high stability, they still face low oxidation reduction potentials and prematurely terminate charging before the battery is completely charged. Consequently, these compounds are electrochemically restricted to low-voltage cathode applications under current technological paradigms [86]. Improvements to the stability of metallocene-based redox shuttle additives are urgently needed to popularize their applications in commercial batteries.

Dimethoxybenzene and its derivatives are typical redox shuttle additives for high-voltage systems, though they have been restricted by their limited solubility and slow mobility. Adachi et al. have systematically explored aromatic compounds possessing methoxy groups with halogens substituted directly on the benzene ring, revealing a two-electron redox shuttle mechanism in the 1 M LiPF_6_ PC/DMC electrolyte, which achieves efficient overcharging protection in a 4 V class battery [87]. However, the abovementioned additives are not entirely effective in preventing overcharging in LiFePO_4_/graphite cells due to the oxidative electro-polymerization between neutral molecules and radical cations. At present, numerous studies are focused on constructing steric hindrance around the benzene ring, predominantly to prevent radical cations from accessing neutral molecules [88]. Hence, the design of the substituents on the benzene ring has endowed dimethoxybenzene-based redox shuttle additives with multi-functional features, such as higher solubility and potential in practical batteries (Figure 8). For example, Zhang et al. synthesized a 2,5−di-tert-butyl-1,4−bis(2-methoxyethoxy)benzene (DBBB) functional redox shuttle with superior solubility in common electrolytes [89]. In the DBBB molecule, the CH_2_-CH_2_O unit can remarkably facilitate the dissolution of lithium salts through strong O−Li^+^ coordination. A cell employing DBBB demonstrates outstanding electrochemical reversibility, withstanding over 180 cycles of 100% overcharge at a 0.5 °C rate. The 1.5 Ah LiFePO_4_/graphite pouch cell also passed standard and abusive overcharging tests, achieving an excellent durability for overcharge protection during a 700-cycle test period without any observable cell swelling. To associate with higher-voltage cathodes, such as LiCoO_2_, LiNi_1/3_Mn_1/3_Co_1/3_O_2,_ and others, Moshurchak et al. produced a 1,4-di-t-butyl-2,5−bis(2,2,2−trifluoroethoxy)benzene (DTFDB) additive with a strong electron-withdrawing 2,2,2−trifluoroethoxy group [90]. As expected, DTFDB exhibited an elevated reversible oxidation potential and maintained a 100% overcharge capacity for each cycle across various cell chemistries. Building on these results, Weng et al. synthesized a novel asymmetric redox shuttle additive, TFDB, by substituting one of the methoxy groups with an OCH_2_CF_3_ group in DTFDB [91]. This additive not only addresses the issue of solubility but also increases the reversible oxidation potential. Generally, phosphonates are employed as flame retardants with high-voltage compatibility in LIBs. Zhang et al. synthesized a tetraethyl-2,5−di-tert-butyl-1,4−phenylene diphosphate (TEDBPDP) additive and investigated its applicability when matched with LiMn_2_O_4_ and Li_1.2_Ni_0.15_Co_0.1_Mn_0.55_O_2_ cathodes [92]. Notably, this redox shuttle additive successfully provided overcharge protection at 4.75 V vs. Li^+^/Li, broadening the design principles for high-potential additives. Huang et al. explored the failure mechanism of the redox shuttle by synthesizing a BTMSDB with a trimethylsilyl group [93]. The substituted trimethylsilyl group is utilized as a chemical probe that can be broken, inducing more prone cation polymerization during the formation of the SEI on the negative electrode surface. This results in an inferior overcharge protection duration for the battery. Weng et al. studied a series of redox shuttle additives by replacing the ditert-butyl groups with 1,2,3,4−tetrahydronaphthalene [94]. This axisymmetric molecule with a rigid skeleton features high solubility in carbonate electrolytes. When the methoxy groups on TDTN were substituted by trifluoroethyl groups, its anti-overcharge potential showed a distinct increase (about 0.34 V higher). A bis-annulated 9,10−bis(2-methoxyethoxy)-1,2,3,4,5,6,7,8−octahydro-1,4,5,8−dimethano-anthracene molecule (BODMA) also exhibited enhanced overcharging prevention capability at a low concentration (0.1–0.2 M) [95]. These findings suggest that symmetry leads to a uniform distribution of excess positive charge between the two methoxy groups, which results in the symmetry of the conformationally locked radical cation, elucidating the markedly increased oxidative stability on the positive side [96].

#### 4.2.2. Electric-Responsive Separators

Commercial separators are typically monolayer or multilayer polymer films and play a key role in impeding electronic circuits and facilitating unrestricted ion migration between positive and negative electrodes [97]. The structural design of a separator dictates the battery’s responsive features. Voltage-responsive separators deliver self-protection against overcharging through their electroactive polymer coating, which results in a reversible phase conversion between the insulating and conductive states. The electroactive polymer on the separator can transition to a conductive state, which induces an internal short circuit, ensuring a harmless overcharge current across the cell.

Chen et al. fabricated poly(3-butylthiophene) (P3BT) featuring a relatively high oxidation potential (3.25 V vs. Li^+^/Li) and stability toward lithium metal [98]. The electronic conductivity of P3BT varies from 10^−9^ to 0.1 S cm^−1^ under various electric states. In the TiS_2_-Li cell with a separator modified by P3BT, a fixed potential of 3.10 V can be maintained, and it does not reach 4.0 V within 12 h, even under overcharge abuse conditions. A P3BT-modified separator has also been applied for safety control in LiFePO_4_||Li_4_Ti_5_O_12_ batteries [99]. The P3BT results in redox-active cycling, which controls overcharging at a low loading of 0.062 mg cm^−2^ in commercial batteries. Ai et al. proposed a polytriphenylamine (PTPAn)-based electroactive polymer as an electric-responsive separator [100]. Utilizing this separator, the LiFePO_4_||Li cell maintains a charging voltage plateau below 4.0 V during the overcharging process. These results reflect the effectiveness of the protection provided by the reversible internal short circuit of the electroactive polymer in the separator. To develop high-energy-density batteries with high-potential cathodes, elevating the oxidation potential of electric-responsive separator is urgently needed. However, single-layer polymer separators generally face challenges in industrial production and commercial applications, especially at a thickness level of dozens of micrometers [101]. Therefore, constructing a bilayer separator has become an approach to meeting the requirements for 4 V level overcharge protection in LIBs. The antioxidant Poly(3-phenylthiophene) (P3PT) for cathodes and the highly stable P3BT for anodes are coated on separators, maintaining stable cycling for LiFePO_4_ cells at 3.9 V (~140 mA h g^−1^) [102]. In a high-nickel ternary cathode (LiNi_0.8_Co_0.15_Al_0.05_O_2_), utilizing an external separator also retained a stable potential of 4.3 V through the external shunt. Ai et al. synthesized a composite separator containing p-polyphenyl (PPP) and polyaniline (PAn) [103]. The LiMn_2_O_4_||Li half-cells within this separator can be cycled within the range of 3.6–4.3 V under overcharge protection conditions. This is because the electroactive polymer undergoes p-doping at high oxidation potentials, entering a conductive state, and reverts to a de-doped at low reduction potentials, isolating state at normal operating voltages, enabling a reversible, voltage-regulated current shunt. At present, there is an urgent need for simpler, more efficient, and less costly fabrication methods for multilayer separators. Ni et al. proposed an integrative strategy for incorporating a reversible silicone-capped electroactive polyfluorene (PFO-PSSQ) into PVDF separators, creating a composite electrolyte [104]. The experiments demonstrated that the interpenetrating polymer electrolyte exhibited higher thermal stability and a lower threshold value than commercial separators when subjected to overcharge abuse, achieving a reversible and stable protection for the LiCoO_2_||Li half-cell even at 4.2 V.

The typical approach to addressing electrical abuse involves establishing a protective voltage that is lower than the peak voltage during overcharging. Electro-polymerization molecules show strong capabilities for interrupting electrochemical reactions, thereby guaranteeing overcharge prevention. However, a significant drawback remains in dealing with the over-discharge process, as electro-polymerization molecules have poor capability for dealing with comprehensive electric abuses. Redox shuttle molecules are another option for electrolyte additives, offering repetitive overcharge protection and favorable cyclic behavior. In general, excessive redox shuttle additives can cause deterioration in electrochemical performance. To ensure its long-term and stable application during the overcharge period, addressing issues such as low solubility and synthesis cost is crucial. In addition, designing a voltage-responsive separator can address these issues via a shunting mechanism. However, its intricate fabrication and assembly process lead to high costs and limited industrial applicability.

### 4.3. Mechanical-Responsive Safety Materials

In operating LIBs, lithium deposition and stripping behaviors are limited by diffusion kinetics, which means that irregular lithium deposits reflect their invalidation states [105]. In general, Li^+^ will become depleted in the electrolyte near the cathodic electrode, breaking the electroneutral balance and forming a local space charge on the surface of the deposition electrode, resulting in the growth of harsh dendrites [106]. Therefore, monitoring the evolution trends of lithium dendrites and implementing protective actions through internal mechanical signals are crucial for designing internal safety components.

#### 4.3.1. Lithium Dendrite Monitoring

In recent years, designed polymer separators have been implemented in LIBs for monitoring lithium dendrites inside the battery. As shown in Figure 9a, Wu et al. fabricated a sandwich-like separator, which integrated a pre-deposited copper layer, around 50 nm in thickness, inside the PE using the magnetron sputtering method (PE-Cu-PE). This separator provides a unique signal in the form of an obvious voltage change between the cathode and anode [107]. When the early lithium dendrites reach the copper layer, the potential shifts to 0 V, allowing for early activation prior to short-circuiting. Lin et al. proposed an integrated porous polyimide/copper/porous polyimide separator (PI/Cu/PI) [108]. This bi-functional separator achieves enhanced thermal stability and is highly sensitive to lithium dendrites (Figure 9b). An alternative approach involves using an active inorganic interlayer for the detection of lithium dendrites. Wang et al. inserted non-conductive red phosphorus (RP) into a bilayer PP separator (Figure 9c) [109]. Once the lithium dendrites penetrate the PP separator, a reaction between lithium and RP can cause a distinct voltage disturbance, avoiding additional electrical risks from conductive electrodes (Cu).

#### 4.3.2. Lithium Morphology Modification

Apart from detecting the growth of lithium dendrites, restraining the effect of separators with the elevated Young’s modulus has become a feasible approach. Lee et al. utilized an ultra-thin copper thin film (CuTF) to enhance the strength of the PE separator membrane and Li utilization during the Li stripping process [110]. This copper coating serves as a current collector for backside plating to manipulate the morphology of lithium metal deposits, ensuring a stable operation without internal short circuits, even with twofold excessive lithium deposition. The nucleation overpotential of lithium metal determines its morphology on the substrate metal [111]. For instance, Ma et al. employed lithiophilic Au nanoparticles on a commercial separator using the magnetron sputtering method [112]. The Au nanoparticles undergo alloying with lithium during the deposition process, forming a Lix-Au alloy before reaching 0 V (vs. Li+/Li), with nearly negligible nucleation potential. Yan et al. demonstrated the regulated behavior of graphite lithiation by the Au layer, which effectively redirects the initial growth of lithium dendrites toward the separator when lithium dendrites emerge during high currents, preventing the growth of potentially hazardous dendrites (Figure 10a) [113]. Recently, alloying reactions have been uncovered between Li metal and various metal substrates, such as Mg, Zn, and Ge. In addition, metallic compounds, such as oxides, nitrides, and fluorides, also exhibit an excellent modification feature for restraining lithium dendrites [114,115,116]. Yan et al. conducted a systemic investigation of a transition metal oxide (TMO) particle coating for PP separators (Figure 10b) [117]. MnO exhibits trace solubility and produces a slow-release effect in the ether electrolyte. In this case, Mn and Li_2_O products are formed through a spontaneous reaction between MnO and Li metal, enabling a “self-repair” effect that fixes the SEI during the lithium metal deposition process. Titanium oxide also presents a functional layer for modifying the mass transfer behavior of Li^+^ due to its electronic localization. Huang et al. observed that TiO_2−x_ shows a strong repulsive force toward Li^+^ through theoretical calculations, facilitating a rapid and even lithium diffusion across the functional separator [118]. This TiO_2−x_@PP separator can improve LIBs’ cyclic performance due to its smooth lithium deposition. Generally, the positively charged oxygen vacancies in TMOs impede the migration of lithium salt anions and prolong the nucleation time of lithium dendrites, ultimately achieving a dendrite-free morphology on the lithium metal anode [119].

However, the poor dissolution of TMOs in organic electrolytes can easily result in an uneven, TM-rich SEI, which is harmful to the coulombic efficiency (CE) [120]. Therefore, TMO coatings in the design of functional separators require more comprehensive investigation. Metal nitrides exhibit excellent stability in organic electrolytes; for example, Li_3_N with a high Li^+^ conductivity can facilitate a steady SEI, achieving a reduction in Li^+^ concentration polarization and lithium dendrites. Yan et al. proposed a Mg_3_N_2_-decorated functional separator that occurs in situ on the lithium metal surface, forming a mixed ion/electron conducting layer (MCL) containing Li_3_N and Li-Mg solid solutions [121]. The elevated Li^+^ conductivity (around 10^−3^ S cm^−1^ at room temperature) in MCL can buffer the Li^+^ concentration gradient and interfacial resistance, thus preventing sustained lithium dendrite growth and side reactions during the cycling process, which extends the battery’s lifespan. As seen in Figure 10c, indium nitride (InN) and aluminum nitride (AlN) also exhibit a cooperative protective effect in LIBs. Ma et al. proposed that the Li||Cu battery utilizing an InN-decorated separator achieves a high CE of over 97% and ultra-stable cycling over 200 cycles [122]. The AlN stands out for its outstanding compatibility with lithium metal due to its impressive thermal conductivity (319 W (m^−1^ K^−1^)), high rigidity (23.7 GPa), and electrochemical stability [123]. As expected, the Li||LiFePO_4_ battery with an AlN-decorated PP separator delivers a high specific capacity of 84.3 mA h g^−1^, even at 10 C. Fluorides play a crucial role in LIBs by boosting flame retardant properties and forming a LiF-rich SEI. Although LiF shows much lower Li^+^ conductivity compared to Li_3_N and Li_2_O, it has a competitive Young’s modulus (around 64.9 GPa) and is effective at preventing electron tunneling while facilitating Li^+^ surface diffusion, thus avoiding lithium dendrite growth [124]. For instance, AlF_3_ and MgF_2_ can form alloy anodes and LiF to protect the lithium metal surface in situ during the lithium deposition process, which enables the battery to have a prolonged cycling life (Figure 10d) [125].

The penetration of rigid lithium dendrites through the solid electrolyte interphase (SEI) induces mechanochemical degradation, while their fracture during prolonged cycling generates electrochemically inactive “dead lithium”. This dual failure mechanism progressively increases interfacial impedance and accelerates capacity fade. Therefore, adaptive smart materials for electrode interfaces have attracted significant attention. Liu et al. utilized silicone polymers to construct “Silly Putty” (SP) as an adaptive protective layer for lithium metal [126]. The SP has a liquid nature that ensures optimal coverage, irrespective of volume fluctuations in the lithium metal under normal conditions, while its shear force increases under mechanical triggers, thereby inhibiting the further growth of lithium dendrites. For the SP layer, the reversible alteration between its ‘‘solid” and ‘‘liquid” features in response to the growth and elimination of lithium dendrites ensures the stable operation of the lithium metal anode.

**Figure 10 polymers-17-01227-f010:**
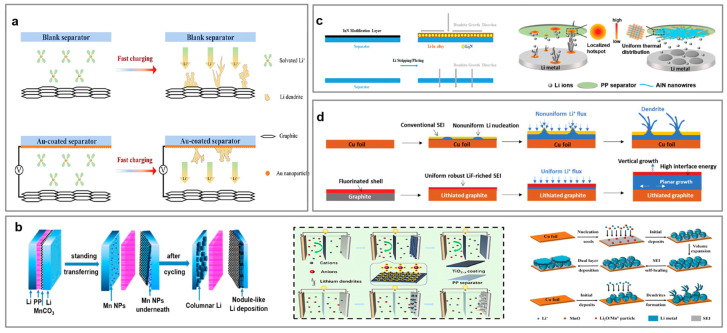
(**a**) The regulating behavior of the functional separator coated with an ultra-thin layer of Au nanoparticle lithium deposition [113] (Copyright 2022 Elsevier); the functional separator with coated TMOs (**b**) [117] (Copyright 2021 Wiley); TMNs (**c**) [122] (Copyright 2022 ACS); and TMFs (**d**) [125] (Copyright 2020 RSC).

These attempts to develop functional separators play a crucial role in suppressing the lithium dendrite formation, ultimately reducing the risk of internal short circuits in LIBs. However, several challenges remain for further investigation: (1) The thickness of the modification layer affects internal impedance. Therefore, constructing a protective layer with even and rapid ion diffusion that meets the requirement for increased charge and discharge rates becomes an important direction for development. (2) It is worth considering the stability of metallic compounds toward lithium metal, as this may inevitably reduce the first coulombic efficiency [127]. (3) Reducing the weight of inorganic coatings on separators is imperative for achieving higher energy density in advanced batteries, which has garnered significant attention.

## 5. Conclusions

This review systematically evaluates four categories of smart safety materials—phase change materials, positive temperature coefficient materials, shape memory materials, and sol-gel transition polymers in response to multi-parameters—and provides detailed strategies for addressing various abuse conditions. Alhough the designed polymeric materials have demonstrated their hierarchical intervention in electrodes, electrolytes, current collectors, separators, etc., for thermal runaway (TR) suppression, the following critical challenges persist in developing high-safety lithium-ion batteries (LIBs): (1) For thermo-responsive systems, the polymeric materials in operating batteries still suffer from a sluggish thermo-responsive window and small “switching ratios”, resulting in poor suppression of thermal runaway propagation. (2) For electric abuse, recent studies have generally focused on overcharge protection in electric-responsive safety materials design. Electro-polymerization and redox shuttle additives still face the intractable problem of molecular synthesis, which restricts their compatibility with high-voltage systems under extended cycles. (3) In operating LIBs, the main strategy focuses on lithium dendrite monitoring and restraining, aiming to achieve excellent cycling properties with a dendrite-free anode. However, the complex fabrication process and additional weight of mechanical-responsive components still impede their practical application in advanced batteries.

## 6. Perspectives

In order to meet the high safety requirements in high-energy-density battery systems, we list several emerging directions for smart safety materials design.

### 6.1. Advanced Characterization Techniques for Detection of Invalidation Status

In order to construct much safer batteries, characterization and early warning techniques have been developed to detect battery invalidation status, though current developments are often insufficient. During the TR process, temperature, voltage, current, and pressure signals change unpredictably. Optical technology, with minimal electromagnetic interference, shows potential for application in high-precision monitoring [128]. Koch et al. conducted a systematic investigation of the characteristic parameters of TR, revealing that gas sensors consistently provide early signals compared to other sensors [129]. Cai et al. also suggested that the gas sensing method achieves the fastest response time (approximately 85 s) for TR using the COMSOL 6.3 model [130]. These innovative simulation and sensor technologies facilitate synchronous and precise measurement of internal invalidation status during the early stages of TR, enabling effective suppression of TR propagation.

This review discusses multiple parameters that can reflect the status of operating batteries, though potential issues still exist in terms of design principles and categories of sensors. Therefore, there is a pressing need to develop effective technologies to mitigate cross-talk between thermal, electric, mechanical, gaseous, and optical signals, achieving accurate sensing and response to disaster chain reactions in batteries.

### 6.2. Cross-Scale Response of Smart Materials for LIBs

Smart materials based on characteristic parameter responses mainly focus on the design of electrodes, electrolytes, separators, current collectors, and other components, enabling swift intervention to interrupt the TR chain reaction at the cell level. However, protection strategies at the single level only offer limited efforts in early TR inhibition. For instance, PTC electrodes can suppress overheating through ionic or electric blocking effects, but the efficiency depends on the status of TR propagation. Notably, the reversibility of thermo-responsive components is also restricted by persistent caloric accumulation.

Therefore, the cross-scale response of smart materials in LIBs needs to integrate the “atom-interface-cell-system” chain, achieving a transition from “passive protection” to “active immunity” through the synergy of material innovation, structural optimization, and intelligent management. Future development will focus on the cross-scale integration of solid-state batteries, AI-driven closed-loop designs, and robustness validation under extreme conditions, laying a safety foundation for the large-scale application of high-energy-density batteries.

### 6.3. Utilization of High-Safety Redundancy Components

In the context of conventional LIBs using liquid electrolytes, the high flammability of batteries remains a key concern in the context of the requirement of high energy density. To address this, cutting-edge battery systems offer two promising solutions: solid-state electrolytes and non-combustible electrolytes.

Solid-state electrolytes mainly consist of inorganic electrolytes (IEs), solid polymer electrolytes (SPEs), and composite solid electrolytes (CSEs). IEs have high ionic conductivity and mechanical strength, enabling highly integrated configurations in battery systems. Despite significant efforts directed toward their development, their properties still lag behind those of liquid electrolytes due to poor interface contact caused by their relatively brittle nature. Benefiting from good flexibility and machinability, SPEs such as PEO or PAN usually exhibit an infiltrative interface. However, they still face the crucial issues of limited strength and ionic conductivity. CSEs are a compromise approach to meet the requirement of practical application by manipulating the composition and structure of inorganic and organic phases, focusing on high ionic conductivity, good interface stability, and considerable mechanical properties. Ideally, solid-state electrolytes should have low electronic conductivity (<10^−10^ S cm^−1^), high Li^+^ conductivity (>10^−3^ S cm^−1^), good ionic transport number, and reliable chemical compatibility, among other mechanical characteristics. Therefore, future research should focus on developing new solid-state electrolyte materials and optimizing compatibility at existing solid–solid interfaces, fully unlocking the conflict between safety and electrochemical properties in solid-state batteries.

In addition, non-combustible electrolytes are an emerging approach to enhance intrinsic safety in LIBs. The utilization of ionic liquids, fluorinated reagents, phosphate esters, or phosphonitriles presents an effective approach to improving safety in high-energy-density batteries. However, numerous issues such as dissolution of lithium salts, low conductivity, and poor interface stability remain. It is imperative to recognize that flammability is not exclusive to liquid electrolytes. When operating batteries are used under high charging states, highly oxidized cathodes and lithiation anodes also pose significant combustion risks. Notably, interfacial reaction design methods offer a means to optimize the compatibility between safety components, thereby enhancing the overall performance of the batteries.

## Figures and Tables

**Figure 1 polymers-17-01227-f001:**
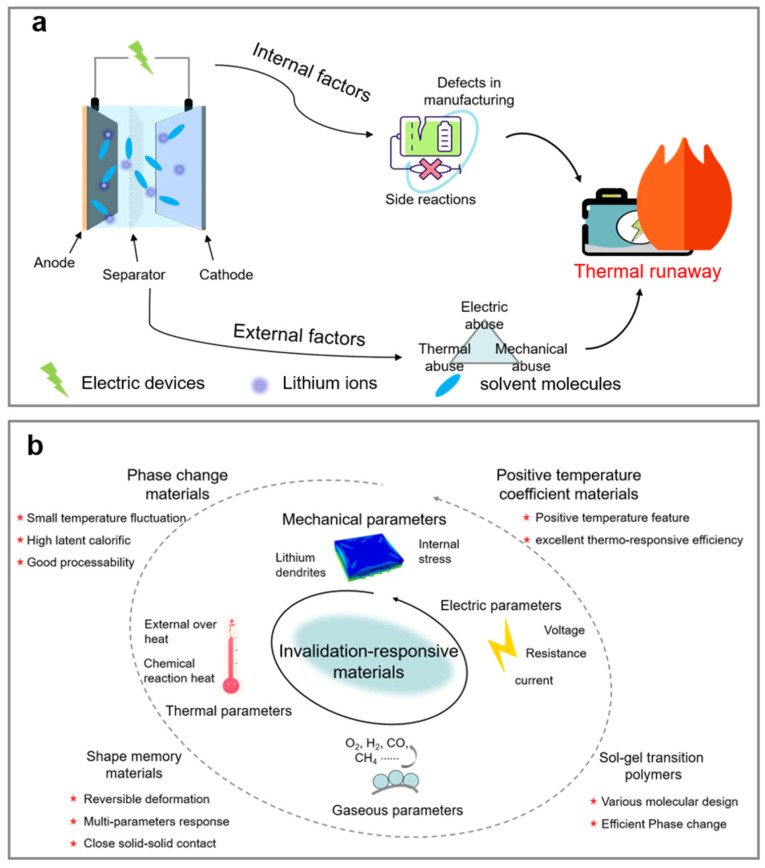
(**a**) The predisposing factors of TR in LIBs; (**b**) the application of invalidation-responsive materials.

**Figure 2 polymers-17-01227-f002:**
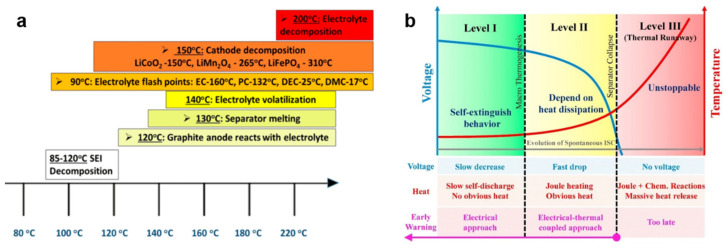
(**a**) Stages of thermal runaway development in LIBs [18] (Copyright 2020 Elsevier); (**b**) sequence of characteristic parameters during thermal runaway process (underlined text displays the evolution of internal short circuit (ISC) and dashed lines reflect the period of TR) [20] (Copyright 2018 Elsevier).

**Figure 4 polymers-17-01227-f004:**
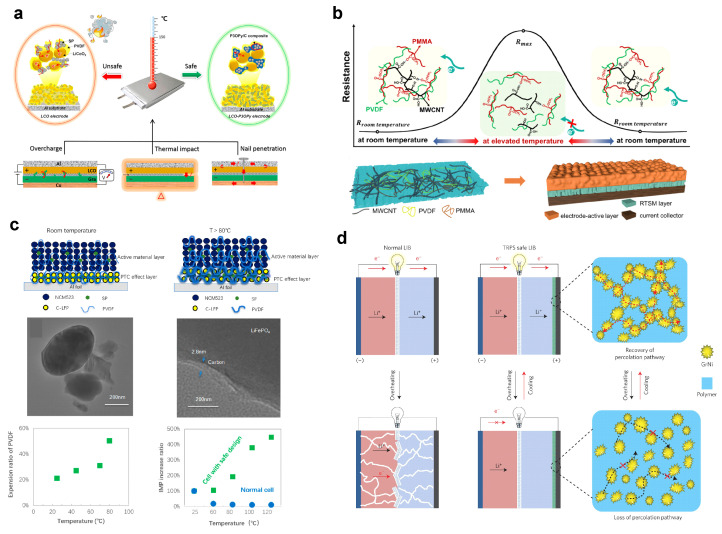
(**a**) Working mechanism of lithium-ion batteries with and without a PTC cathode under thermal abuse condition [38] (Copyright 2019 Elsevier); (**b**) schematic illustration of the structural features and temperature-responsive mechanism of the as-proposed RTSM composite and its role between the electrode-active layer and the current collector [39] (Copyright 2022 ACS); (**c**) schematic diagram of a cathode electrode with a PTC layer safe design [40] (Copyright 2021 ACS); (**d**) schematic illustration of safe battery design [43] (Copyright 2024 Nature).

**Figure 5 polymers-17-01227-f005:**
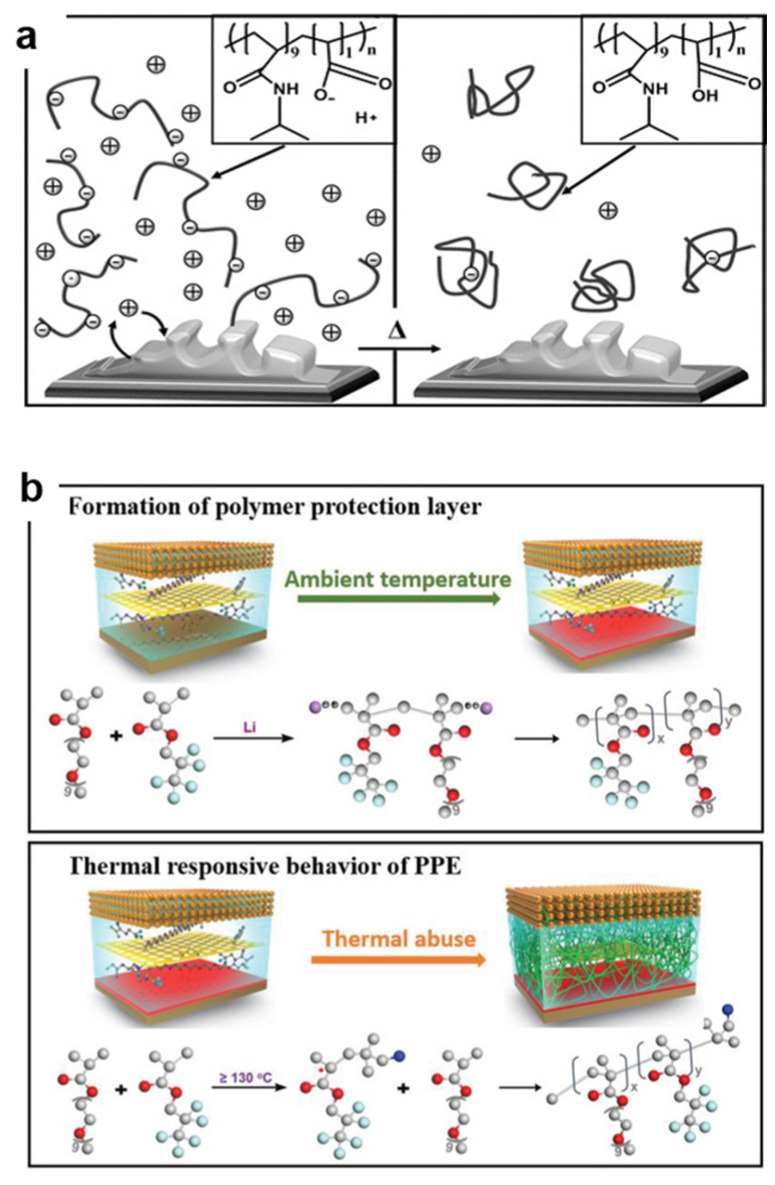
(**a**) The ionic blocking effect of a thermally controllable polymerelectrolyte for electrochemical energy storage [45] (Copyright 2012 Wiley); (**b**) schematic illustration of the formation of the polymer protection layer and its smart thermal-responsive behavior in the cell by the free radical polymerization mechanism under thermal abuse conditions [48] (Copyright 2020 Wiley).

**Figure 6 polymers-17-01227-f006:**
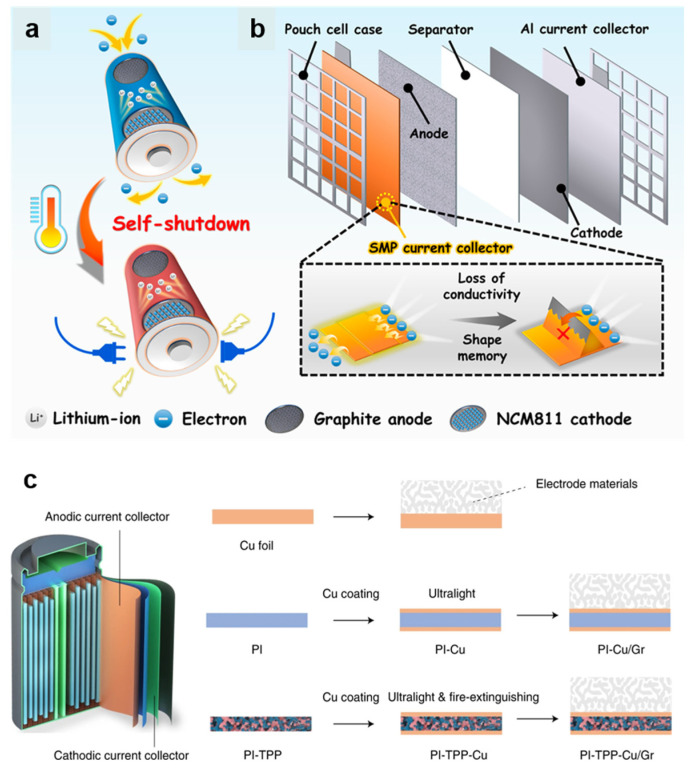
Design of the micropatterned and shape-memorized current collectorsL (**a**) self-shutdown performance of safe lithium-ion batteries with smart current collectors before thermal runaway [49] (Copyright 2022 ACS); (**b**) the safe battery internal structure and the trigger mechanism of the automatic cut-out current collector [49] (Copyright 2022 ACS); (**c**) conventional pure Cu CCs are heavy and bulky, while PI-Cu CCs are much lighter. By incorporating TPP and subsequently coating the CC with ultra-thin Cu layers on both sides, the resulting PI-TPP-Cu CC is ultralight and exhibits efficient flame retardant properties [50] (Copyright 2020 Nature).

**Figure 7 polymers-17-01227-f007:**
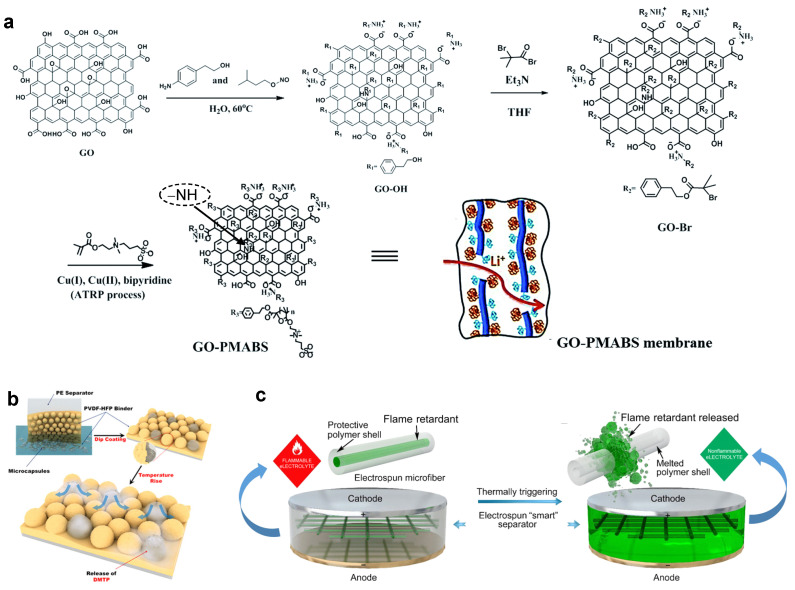
(**a**) Synthesis of graphene oxide functionalized with poly(3-(N-2-methacryloyloxyethyl-N,N-dimethyl)ammonatobutane-sulfonate, GO-PMABS [57] (Copyright 2014 RSC); (**b**) schematic illustrations of the synthesis route of microcapsules containing the fire-suppression agent (DMTP) [62] (Copyright 2015 ACS) (**c**); schematic of the “smart” electrospun separator with thermal-triggered flame-retardant properties for lithium-ion batteries [63] (Copyright 2017 Science).

**Figure 8 polymers-17-01227-f008:**
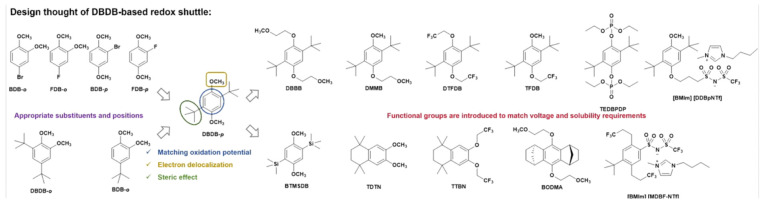
Dimethoxybenzene and its derivatives as redox shuttles: a summary of the design logic [90] (Copyright 2024 Elsevier).

**Figure 9 polymers-17-01227-f009:**
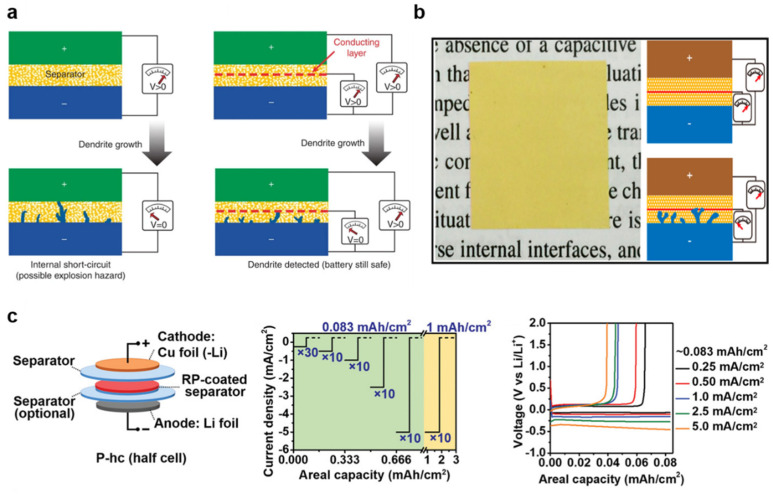
(**a**) Dendrite formation in a traditional lithium battery and the functional separator, which can detect early lithium dendrite penetration through an internal short circuit when the *V*_Li-Li_ drops to zero [107] (Copyright 2014 Nature); (**b**) the PI−based smart separator and its detection mechanism for early lithium dendrites [108] (Copyright 2016 ACS); (**c**) configuration of the half−cell with RP coating (P−hc) and its testing protocol, showing the current density versus time [109] (Copyright 2019 Wiley).

**Table 1 polymers-17-01227-t001:** Typical LIB accidents at home and abroad from 2023 to 2024.

Time	Type	Consequence
April 2023	A lithium battery container caught fire at an industrial park in Gothenburg, Sweden	Massive property damage
May 2023	A 5MW energy storage facility caught fire in East Hampton, New York, USA	Massive property damage
June 2023	A fire broke out at an electric bicycle shop in Chinatown, New York City	Four people died, and two were seriously injured
July 2023	A fire broke out at the container energy storage station in Longjing District, Taichung City, Taiwan Province	Massive property damage
August 2023	A storage energy cabinet suddenly caught fire at the Guangtong Logistics Park in Zhuhai, Guangdong Province	Massive property damage
August 2023	Lithium battery failure in an electric scooter caused a fire in a residential building in Los Angeles, California	Two people died, and multiple people were seriously injured
September 2023	A fire broke out in an apartment building due to overheating lithium batteries in personal mobility devices in London, UK	Significant property damage, and many people received treatment for smoke inhalation
February 2024	A lithium-ion battery from an electric bicycle caused an apartment fire in the Harlem neighborhood of New York City	One journalist died, and multiple people were seriously injured
May 2024	A fire broke out at a 70 MW agricultural-photovoltaic complementary energy storage power station in Hainan Province	A group of prefabricated battery containers was burned

**Table 2 polymers-17-01227-t002:** The definition of thermal runaway and propagation.

Number	Standard	Description
1	GB/T 36276-2023 [14]	The phenomenon of uncontrollable temperature rise caused by exothermic reactions inside the battery cell
2	IEC 62619-2022 [15]	Uncontrollable and rapid temperature rise caused by exothermic reactions within the battery cell
3	UL 1973–2022 [16] UL 9540A-2023 [17]	An event in which an electrochemical battery uncontrollably raises its temperature through self-heating. Thermal runaway occurs when the heat generated by the battery exceeds the heat it can dissipate. This can lead to fires, explosions, and gas emissions
4	GB/T 36276-2023	The phenomenon where thermal runaway in a battery cell within a battery module triggers thermal runaway in adjacent or other cells

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
