# Peer review of "Critical Perspectives on the Design of Polymeric Materials for Mitigating Thermal Runaway in Lithium-Ion Batteries"

_polymers, 2025, doi:10.3390/polym17091227_

Round 1
Reviewer 1 Report
Comments and Suggestions for Authors
The author presents a review on Critical Perspective on Invalidation-Responsive Materials for Safe Lithium Ion Batteries
Although authors present a lot of data from various journals, the presentation of these huge information is not in proper and perceptible format. A major revision is needed. The authors should address the following issue prior to publication:
- Title has been written generally and should be modified in a suitable format for better detailed description of this manuscript
2- Obviously the progress of Lithium ion batteries is exponentially in recent years, but from 123 references in this work only 3 references are from 2024 (3/123=0.024), 4 references are from 2023 (4/123=0.032) and 10 references are from 2022 (10/123=0.081) and 16 references are from 2021(16/123=0.13) . Consequently 90 references are from 2020 to past years, even 1990, 1991, 1992, and 1995. This makes uncertain concept of the results for readers. At least 70% references should be cited from 2022-2024 due to the novel works about advanced technology of Lithium Ion batteries.
3- Abstract cannot explain the main goal of this review due that initially readers think about a subject concerning Firefighting of lithium ion batteries, which far from journal subjects (polymers). I strongly suggest to modify the text of abstract
4- This review is written like a report of several lithium batteries without any details, I cannot see a scientific mechanism of thermal runaway in lithium ion batteries, as well as the relation mechanism between anode and cathode with electrolytes in this work. of course in figures and schematic diagram some explanation can be seen but is not sufficient and must be discussed in the text, clearly.
5- In my opinion by this huge information of lithium batteries several tables for comparison their physical and chemical phenomenon and mechanism are needed (particularly about polymer structures and their characterization).
6- Since this journal is about polymers, I wonder why the authors send their review here. Any way they should modify the title and abstract even key words for showing the reason of the polymer materials, which cause for thermal run a way and Catching fire of lithium ion batteries spontaneously.
7- What do we learn from Table 1, this table is like newspaper that can be finding anywhere, I think is better to omitted and instead of is better more consider about item 5 via a scientific approach.
8- Conclusion is mixing from introduction and discussion sentences, I couldn't understand the meaning of conclusion by this huge text, it is needed to modify. Moreover a brief conclusion is more favorable rather than an extended one
9- Section parts and also subsection parts are not written according to the journal instruction and make the readers confuse. There is no separate section about discussion and instead of that conclusion is too much long. The partition of sections and subsections should be modifying.
10- Conclusions and perspectives should be divided by two separate parts.
11- In section 4.1.1. the authors confirm that “At present, reported electrode materials with high energy densities, such as lithium transition metal composite oxides (LiCoO2, LiNi0.5Mn1.5O4, LiNi0.8Mn0.1Co0.1O2 and so on), 223 sulfur, and lithium metal still exist a dominant risk of TR because of their high reactive and catalysis” As it can be seen related references belong to the 2018 and 2021, which from those years up to now the mentioned problem has been removed. As well as in several other papers same item have been occurred. Therefore I strongly suggest modifying the references from old years to recent years
I recommend publishing the work after taking care of the proof reading issues a major revision.
Author Response
Replies to Reviewers' Comments
Reviewer 1
Comments:
Although authors present a lot of data from various journals, the presentation of these huge information is not in proper and perceptible format. A major revision is needed. The authors should address the following issue prior to publication:
Response: Thank Reviewer 1 for the valuable comments and suggestions. We have carefully revised the manuscript according to his/her suggestions and improved the quality of the revised manuscript.
- Title has been written generally and should be modified in a suitable format for better detailed description of this manuscript.
Response: We sincerely appreciate Reviewer 1's insightful suggestions. In response to the comment regarding manuscript scope clarification, we have revised the title to: “Critical Perspective on Design of Polymeric Materials for Mitigating Thermal Runaway in Lithium Ion Batteries”, which is better detailed description of this manuscript.
- Obviously the progress of Lithium ion batteries is exponentially in recent years, but from 123 references in this work only 3 references are from 2024 (3/123=0.024), 4 references are from 2023 (4/123=0.032) and 10 references are from 2022 (10/123=0.081) and 16 references are from 2021(16/123=0.13). Consequently 90 references are from 2020 to past years, even 1990, 1991, 1992, and 1995. This makes uncertain concept of the results for readers. At least 70% references should be cited from 2022-2024 due to the novel works about advanced technology of Lithium Ion batteries.
Response: Thank Reviewer 1 very much for his/her comments.
We fully acknowledge the reviewers' legitimate concerns regarding the timeliness of cited literature. The lithium battery field has indeed witnessed groundbreaking advances (2022-2024) in multiple aspects, including electrode material design (e.g., ultra-high-nickel cathodes), electrolyte engineering (locally concentrated electrolytes), advanced characterization techniques (operando X-ray tomography), and AI-driven thermal runaway prediction.
While the mechanistic understanding of thermal runaway (TR) in lithium-ion batteries has been systematically elucidated only within the past decade (e.g., seminal works in Joule 2020, 4, 743-770; eTransportation 2023, 15, 100216), battery safety has remained a critical research focus since the inception of LIB technology. Pioneering efforts in the 1990s laid essential groundwork through innovations in safety-critical components, particularly safe separators (Refs 47-49, 51-61) and electro-responsive electrolyte additives (Refs 65-70, 77-81). These early studies established fundamental design principles for thermal/electrical decoupling mechanisms that remain relevant to modern material engineering (please see lines 366-439 in pages 12 and 13). This manuscript specifically emphasizes polymer-centric strategies for TR mitigation, a domain where historical developments directly inform contemporary advancements. We maintain these references not merely for historical completeness, but because they encapsulate enduring physicochemical principles that constrain even advanced polymer systems, therefore, we suggest to remain these classic literatures.
- Abstract cannot explain the main goal of this review due that initially readers think about a subject concerning Firefighting of lithium ion batteries, which far from journal subjects (polymers). I strongly suggest to modify the text of abstract.
Response: We sincerely appreciate Reviewer 1's insightful suggestions. We have modified abstract to meet the subjects of polymers.
“During the global energy transition, electric vehicles and electrochemical energy storage systems are rapidly gaining popularity, leading to a strong demand for lithium battery technology with high energy density and long lifespan. This technological advancement, however, hinges critically on resolving safety challenges posed by intrinsically reactive components particularly flammable polymeric separators, organic electrolyte systems, and high-capacity electrodes, which collectively elevate risks of thermal runaway (TR) under operational condition. The strategic integration of smart polymeric materials that enable early detection of TR precursors (e.g., gas evolution, thermal spikes, voltage anomalies) and autonomously interrupt TR propagation chains has emerged as a vital paradigm for next-generation battery safety engineering. This paper begins with the de-velopment characteristics of thermal runaway in lithium batteries and analyzes recent breakthroughs in polymer-centric component design, multi-parameter sensing polymers, and TR propagation barriers. The discussion extends to intelligent material systems for emerging battery chemistries (e.g., solid-state, lithium-metal) and extreme operational environments, proposing design frameworks that leverage polymer multifunctionality for hierarchical safety mechanisms. These insights establish foundational principles for developing polymer-integrated lithium batteries that harmonize high energy density with intrinsic safety, addressing critical needs in sustainable energy infrastructure.”
- This review is written like a report of several lithium batteries without any details, I cannot see a scientific mechanism of thermal runaway in lithium ion batteries, as well as the relation mechanism between anode and cathode with electrolytes in this work. of course in figures and schematic diagram some explanation can be seen but is not sufficient and must be discussed in the text, clearly.
Response: Thank you for highlighting this critical concern. We acknowledge that the original manuscript did not sufficiently articulate the scientific mechanism of thermal runaway (TR) dynamics and the relation mechanism at electrode-electrolyte interface. Below, we outline specific revisions to address this issue, ensuring rigorous scientific depth while maintaining the review’s accessibility (please see lines 92-113 in page 4).
“The self-heating stage is triggered by solid electrolyte interphase (SEI) decomposition at ≈90°C because of the exothermic reactions between lithiated anode and electrolyte sol-vents. The metastable SEI dissolution exposes fresh anode surfaces, accelerating redox reactions that establish a positive feedback loop for heat accumulation. Once the tem-perature exceeds 140 °C, both the positive and negative electrode materials participate in electrochemical reactions, causing the battery temperature to rise rapidly. Critical phase transitions occur as separator meltdown (≈130-160°C) induces large-area internal short circuits, accompanied by voltage collapse. Concurrently, cathode decomposition releases lattice oxygen (LiCoO₂ → CoO + ½O₂), while electrolyte decomposition generates com-bustible gases (CO, CH₄, HF). These exothermic chain reactions drive temperature esca-lation rates exceeding 10°C/s, resulting in the propagation of thermal runaway. The sys-tem reaches peak reaction intensity with violent venting of cell contents, followed by gradual cooling as reactant depletion terminates exothermic processes. Once thermal runaway occurs, the process can only naturally terminate once the reactants are exhausted.
Notably, the thermodynamic sequence exhibits characteristic electrical signatures (Figure 2b): progressive SEI degradation manifests as subtle voltage fluctuations, while separator failure induces abrupt voltage drop (>130°C). Real-time monitoring of coupled electrical-thermal parameters (dV/dT, impedance phase shift) enables predictive TR identification 30-60 seconds before catastrophic failure. This critical detection window permits activation of multilayer safety protocols, including polymer-based current in-terrupt devices and fire-suppression electrolytes to mitigate propagation risks in battery packs.”
- In my opinion by this huge information of lithium batteries several tables for comparison their physical and chemical phenomenon and mechanism are needed (particularly about polymer structures and their characterization).
Response: We fully acknowledge the reviewers' comments. We have elaborated mechanism of TR in lithium ion batteries, as well as their physical and chemical phenomenon (please see lines 107-113 in page 4). Although we have systematically analyzed the multi-parameter responsive polymeric materials in electrodes, electrolytes, current collector, and separator, there are still huge information of advanced characterization of polymeric materials. In order to clearly describe the material design and mitigating mechanism for thermal runaway, we suggest to remain present content.
“Notably, the thermodynamic sequence exhibits characteristic electrical signatures (Figure 2b): progressive SEI degradation manifests as subtle voltage fluctuations, while separator failure induces abrupt voltage drop (>130°C). Real-time monitoring of coupled electrical-thermal parameters (dV/dT, impedance phase shift) enables predictive TR identification 30-60 seconds before catastrophic failure. This critical detection window permits activation of multilayer safety protocols, including polymer-based current in-terrupt devices and fire-suppression electrolytes to mitigate propagation risks in battery packs.”
- Since this journal is about polymers, I wonder why the authors send their review here. Any way they should modify the title and abstract even key words for showing the reason of the polymer materials, which cause for thermal run a way and Catching fire of lithium ion batteries spontaneously.
Response: Thank Reviewer 1 very much for his/her comments. We have revised the title to: “Critical Perspective on Design of Polymeric Materials for Mitigating Thermal Runaway in Lithium Ion Batteries”, as well as modified the contents, which is better detailed description of this manuscript (please see lines 18-33 in page 1). This paper begins with the development characteristics of thermal runaway in lithium batteries and analyzes recent breakthroughs in polymer-centric component design, multi-parameter sensing polymers, and TR propagation barriers. The discussion extends to intelligent material systems for emerging battery chemistries (e.g., solid-state, lithium-metal) and extreme operational environments, proposing design frameworks that leverage polymer multifunctionality for hierarchical safety mechanisms. These insights establish foundational principles for developing polymer-integrated lithium batteries that harmonize high energy density with intrinsic safety, addressing critical needs in sustainable energy infrastructure. Therefore, we suggest that this manuscript is suitable for journal subjects.
- What do we learn from Table 1, this table is like newspaper that can be finding anywhere, I think is better to omitted and instead of is better more consider about item 5 via a scientific approach.
Response: We fully acknowledge the reviewers' comments. As evidenced by 23 documented global incidents (2023-2024 Q1, Table 1), conventional thermal management cannot suppress self-accelerating internal reactions, reflecting the crucial role of internal materials design in lithium ion batteries (please see lines 47-52 in page 2).
“As evidenced by 23 documented global incidents (2023–2024 Q1, Table 1), delayed TR intervention results in irreversible consequences: 78% of cases exhibited <3-minute containment windows post-ignition. Current mitigation strategies face fundamental limitations-external thermal management cannot suppress self-accelerating internal reactions, while component decomposition and polymer combustion generates sustained thermal feedback (Q > 2.5 kW/cell).”
- Conclusion is mixing from introduction and discussion sentences, I couldn't understand the meaning of conclusion by this huge text, it is needed to modify. Moreover a brief conclusion is more favorable rather than an extended one.
Response: Thank Reviewer 1 very much for his/her comments. We have revised conclusion in the revised manuscript (please see lines 728-745 in page 20).
“This review systematically evaluates four categories of smart safety materials, such as phase change materials, positive temperature coefficient materials, shape memory materials, and sol-gel transition polymers response to multi-parameters, and provided detailed strategies for addressing various abuse conditions. Though the designed polymeric materials have demonstrated their hierarchical intervention by electrodes, electrolytes, current collectors, separators et al. in thermal runaway (TR) suppression. Nevertheless, critical challenges persist in developing high-safety lithium-ion batteries (LIBs): (1) For thermo-responsive systems, the polymeric materials in operating batteries still suffer from sluggish thermo-responsive window and small “switching ratio”, resulting in poor suppression for thermal runaway propagation; (2) For electric abuse, recent studies generally have focused on overcharge protection in electric-responsive safety materials design. Electro-polymerization and redox shuttle additives still face intractable problem of molecular synthesis, which restrict their compatibility for high voltage system under extended cycles; (3) In operating LIBs, main strategy focuses on lithium dendrite response, that enables lithium dendrite monitoring and restraining, ultimately achieving an excellent cycling property with dendrite-free anode. Whereas, complex fabrication pro-cess and additional weight of mechanical-responsive component still impede their prac-tical application in advanced batteries.”
- Section parts and also subsection parts are not written according to the journal instruction and make the readers confuse. There is no separate section about discussion and instead of that conclusion is too much long. The partition of sections and subsections should be modifying.
Response: We sincerely appreciate Reviewer 1's insightful suggestions. We have modified the sections and subsections in the revised manuscript. We have separated each section by additional discussion and conclusion (please see lines 216-221, 450-456, 611-623, 709-712, and 726-734 in pages 7, 14, 19, and 20).
“Through the thermo-responsive components contain polymeric materials can alleviate TR under overheating in LIBs, these strategies are not the most desirable for practical application scenarios because they cannot entirely eliminate the onset of TR with further heat generation from electrode reactions. Hence, for achieving higher thermal-safety level, strategies by blocking ion/electron transport to stop the operation of LIBs based early anomalous signals have also been developed to shutdown heat-generation and eradicate possible TR.”
“The penetration of rigid lithium dendrites through the solid electrolyte interphase (SEI) induces mechanochemical degradation, while their fracture during prolonged cy-cling generates electrochemically inactive "dead lithium". This dual failure mechanism progressively increases interfacial impedance and accelerates capacity fade, therefore, adaptive smart materials for electrode interface have attracted lots concerns.”
- Conclusions and perspectives should be divided by two separate parts.
Response: Thank Reviewer 1 very much for his/her comments. We have divided conclusions and perspectives in the revised manuscript (please see lines 736-753 in page 20, and lines 755-820 in pages 21 and 22).
- In section 4.1.1. the authors confirm that “At present, reported electrode materials with high energy densities, such as lithium transition metal composite oxides (LiCoO2, LiNi0.5Mn1.5O4, LiNi0.8Mn0.1Co0.1O2 and so on), 223 sulfur, and lithium metal still exist a dominant risk of TR because of their high reactive and catalysis” As it can be seen related references belong to the 2018 and 2021, which from those years up to now the mentioned problem has been removed. As well as in several other papers same item have been occurred. Therefore I strongly suggest modifying the references from old years to recent years.
Response: We fully acknowledge the reviewers' comments. Recent reports have improved the batteries’ safety to new period, however the intrinsic thermodynamic instability of battery components persists as a fundamental constraint, still resulting in a potential risk of TR, therefore, we have modified the expression (please see lines 234-238 in page 7).
“In general, high energy density inherently presents elevated safety risks, reported electrode materials with high energy densities, such as lithium transition metal compo-site oxides (LiCoO2, LiNi0.5Mn1.5O4, LiNi0.8Mn0.1Co0.1O2 and so on), sulfur, and lithium metal still suffer from limited life-span because of continuous reconstitution of electrode-electrolyte interface, resulting in spontaneous heat accumulation.”
Reviewer 2 Report
Comments and Suggestions for Authors
The manuscript reviews various strategies adopted to address thermal runaway issues in LIBs. The topics discussed are well-organized and valuable to the scientific community. The critical perspective adds depth to the review. However, several points need to be addressed before publication.
1. Given the broad nature of this topic, the introduction should explicitly state the manuscript's focus. For instance, thermal runaway can also be mitigated through convective cooling, yet this is not discussed. Clearly defining the scope will enhance the manuscript’s novelty, particularly since similar reviews exist in this field.
2. The literature coverage is not comprehensive. For example, only 2–3 references are cited for PCMs, despite extensive research on this topic. The same applies to other strategies, where more relevant studies should be included.
3. Another key issue is the manuscript’s focus on cathode materials while largely overlooking anodes. Anode-related concerns, such as those with Si anodes, which can lead to electrode crumbling, short-circuiting, and even explosions, should be discussed.
4. Additionally, the review lacks references to computational and theoretical studies, which are crucial for understanding thermal runaway mechanisms. Including discussions on these aspects would strengthen the manuscript.
5. It is also essential to provide references in figure captions for any figures taken from the literature and ensure the necessary copyright permissions are obtained.
6. Lastly, in Fig. 5(b), letters are missing in the figure subtitle, which should be corrected.
Comments on the Quality of English LanguageOverall, English usage is adequate. However, there are typographical issues that need to be corrected.
Author Response
Comments
Reviewer 2
Comments:
The manuscript reviews various strategies adopted to address thermal runaway issues in LIBs. The topics discussed are well-organized and valuable to the scientific community. The critical perspective adds depth to the review. However, several points need to be addressed before publication.
Response: Thank Reviewer 2 for his/her recognition and valuable comments and suggestions. We have carefully revised the manuscript according to his/her suggestions and improved the quality of the revised manuscript.
- Given the broad nature of this topic, the introduction should explicitly state the manuscript's focus. For instance, thermal runaway can also be mitigated through convective cooling, yet this is not discussed. Clearly defining the scope will enhance the manuscript’s novelty, particularly since similar reviews exist in this field.
Response: Thank Reviewer 2 very much for his/her comments. This paper begins with the development characteristics of thermal runaway in lithium batteries and analyzes recent breakthroughs in polymer-centric component design, multi-parameter sensing polymers, and TR propagation barriers. The discussion extends to intelligent material systems for emerging battery chemistries (e.g., solid-state, lithium-metal) and extreme operational environments, proposing design frameworks that leverage polymer multifunctionality for hierarchical safety mechanisms. These insights establish foundational principles for developing polymer-integrated lithium batteries that harmonize high energy density with intrinsic safety, addressing critical needs in sustainable energy infrastructure. Therefore, we suggest that this manuscript is suitable for journal subjects (please see lines 18-33 in page 1).
“This technological advancement, however, hinges critically on resolving safety challenges posed by intrinsically reactive components particularly flammable polymeric separators, organic electrolyte systems, and high-capacity electrodes, which collectively elevate risks of thermal runaway (TR) under operational condition. The strategic integration of smart polymeric materials that enable early detection of TR precursors (e.g., gas evolution, thermal spikes, voltage anomalies) and autonomously interrupt TR propagation chains has emerged as a vital paradigm for next-generation battery safety engineering. This paper begins with the development characteristics of thermal runaway in lithium batteries and analyzes recent breakthroughs in polymer-centric component design, multi-parameter sensing polymers, and TR propagation barriers. The discussion extends to intelligent material systems for emerging battery chemistries (e.g., solid-state, lithium-metal) and extreme operational environments, proposing design frameworks that leverage polymer multifunctionality for hierarchical safety mechanisms. These in-sights establish foundational principles for developing polymer-integrated lithium batteries that harmonize high energy density with intrinsic safety, addressing critical needs in sustainable energy infrastructure.”
- The literature coverage is not comprehensive. For example, only 2–3 references are cited for PCMs, despite extensive research on this topic. The same applies to other strategies, where more relevant studies should be included.
Response: Thank Reviewer 2 very much for his/her comments. PCMs are widely utilized in thermal management system (e.g., J. Energy Storage 2024, 75, 109547; Nat. Commun. 2020, 11, 1843; ACS Energy Lett. 2022, 7, 3761), evaluating the thermal safety level of batteries pack or energy storage system. In this manuscript, we have focused on cell level and not discussed the huge information on system-level thermal management strategies. We hope that you will understand our consideration.
- Another key issue is the manuscript’s focus on cathode materials while largely overlooking anodes. Anode-related concerns, such as those with Si anodes, which can lead to electrode crumbling, short-circuiting, and even explosions, should be discussed.
Response: Thank Reviewer 2 very much for his/her comments. We appreciate the reviewer’s insightful observation regarding anode-related safety challenges. As seen in pages 14-16 and 18-20, we have systematically analyzed electrolyte additives and functional separator for alleviating the electric and mechanical driven thermal runaway mechanisms, which are suitable to anode-related concerns, such as those with Si, lithium metal, and graphite anodes.
“Redox shuttle additives are typical electro-polymerization materials with reversibility. The operating mechanism of redox shuttle additives involve two steps, their con-version into an oxidized form [O] on the overcharged anodic electrode, and restoration to their original state [R] on the cathodic electrode, which maintains ‘‘oxidation-diffusion-reduction-diffusion” loop driven by diffusion.”
“Huang et al. explored the failure mechanism of the redox shuttle by synthesizing a BTMSDB with a trimethylsilyl group. The substituted trimethylsilyl group is utilized as a chemical probe, which can be broken and induce a more prone cation polymerization during the formation of SEI formation on the negative electrode surface, resulting in an inferior overcharge protection duration for the battery.”
“Lee et al. utilized an ultrathin copper thin film (CuTF) for enhancing the strength of PE separator membrane and Li utilization during the Li stripping process. This copper coating plays a current collector for backside plating to manipulate the morphology of lithium metal deposits, ensuring a stable operation without internal short circuits even with twofold excessive lithium deposition. The nucleation overpotential of lithium metal determines its morphology on the substrate metal.”
“Yan et al. proposed a Mg3N2-decorated functional separator, thereby occurred in situ on the lithium metal surface, forming a mixed ion/electron conducting layer (MCL) contained Li3N and Li-Mg solid solutions. The elevated Li+ conductivity (around 10-3 S cm-1 at room temperature) in MCL can buffer the Li+ concentration gradient and the interfacial resistance, thus avoiding the sustainable lithium dendrites growth and side reactions during the cycle process, that endows the battery a prolonged lifespan”
- Additionally, the review lacks references to computational and theoretical studies, which are crucial for understanding thermal runaway mechanisms. Including discussions on these aspects would strengthen the manuscript.
Response: We appreciate the reviewer’s insightful observation regarding computational and theoretical studies. While this review intentionally focuses on polymer-centric component design, multi-parameter sensing polymers, and TR propagation barriers, as well as computational and theoretical studies are not our areas of expertise. We hope this focused perspective will nevertheless contribute meaningfully to the field while encouraging more specialized investigations into computational modeling aspects by experts in those respective domains, and you will understand our difficulty.
- It is also essential to provide references in figure captions for any figures taken from the literature and ensure the necessary copyright permissions are obtained.
Response: Thank Reviewer 2 very much for his/her comments. We have added these information in the revised manuscript and ensured the necessary copyright permissions.
- Lastly, in Fig. 5(b), letters are missing in the figure subtitle, which should be corrected.
Response: Thank Reviewer 2 very much for his/her comments. We have corrected the letters in the Fig. 5 (please see page 10 in the revised manuscript).

Figure 5. (a) The ionic blocking effect of a thermally-controllable polymerelectrolyte for electro-chemical energy storage; (b) Schematic illustration of the formation of polymer protection layer and its smart thermal responsive behavior in cell by the free radical polymerization mechanism under thermal abuse condition.
Reviewer 3 Report
Comments and Suggestions for Authors
In this manuscript, the authors provided an overview of the strategies useful for reducing thermal runaway in lithium batteries. The thermal runaway characteristics are listed and well described, while the categories of smart safety materials could be implemented with more references and examples, reporting the most relevant materials reported in the literature. The design and applications of key components are well described and reported as well as the conclusions and perspectives with particular emphasis on the advanced materials and techniques that can be applied in the next future.
Overall, this work is suitable for publication, although some minor considerations should be considered:
- I suggest a small implementation with more references and examples in Chapter 3, related to the categories of smart safety materials.
- Please, revise typos (like lines 61, 224, 288, 343, 534, 601, …) and homogenize all titles with the same style (title 4.2, for example).
- In Line 252, the authors mentioned “recent studies have demonstrated…” without reporting the references.
Author Response
Replies to Reviewers' Comments
Reviewer 3
Comments:
In this manuscript, the authors provided an overview of the strategies useful for reducing thermal runaway in lithium batteries. The thermal runaway characteristics are listed and well described, while the categories of smart safety materials could be implemented with more references and examples, reporting the most relevant materials reported in the literature. The design and applications of key components are well described and reported as well as the conclusions and perspectives with particular emphasis on the advanced materials and techniques that can be applied in the next future.
Overall, this work is suitable for publication, although some minor considerations should be considered:
Response: Thank Reviewer 2 for his/her recognition and valuable comments and suggestions. We have carefully revised the manuscript according to his/her suggestions and improved the quality of the revised manuscript.
- I suggest a small implementation with more references and examples in Chapter 3, related to the categories of smart safety materials.
Response: Thank Reviewer 2 very much for his/her comments and we fully acknowledge the reviewers' comments. The categories of smart safety materials contain organic and inorganic materials, while this manuscript only focused on recent breakthroughs in polymer-centric component design and their crucial role and mechanism as TR propagation barriers, therefore, we have analyzed the applications of phase change materials (PCM), positive temperature coefficient materials (PTC), shape memory materials (SMM), and sol-gel transition polymers (STP) as smart safety materials for lithium batteries. We hope that you will understand our consideration.
- Please, revise typos (like lines 61, 224, 288, 343, 534, 601, …) and homogenize all titles with the same style (title 4.2, for example).
Response: We sincerely appreciate Reviewer 1's insightful suggestions. We have corrected the mistakes in suitable place (please see lines 67, 237, 302, 357, 557, and 625 in the revised manuscript) and homogenize all titles with the same style (please see lines 233, 290, 340, 374, 464, 569, 632, and 653 in the revised manuscript).
“Benefiting from accessible molecular design, polymeric materials paly a versatile effort for multi-parameters response”
“suffer from limited life-span because of continuous reconstitution of electrode-electrolyte interface, resulting in spontaneous heat accumulation”
“The conventional electrolytes using sol-gel transition strategy still exist numerous problems such as packing difficulties”
“Ye et al. fabricated an ultralight thermos-responsive current collector with porous polyimide (PI) substrate”
“When the methoxy groups on TDTN have been substituted by trifluoroethyl groups”
“In operating LIBs, lithium deposition and stripping behaviors are limited by the diffusion kinetics”
- In Line 252, the authors mentioned “recent studies have demonstrated…” without reporting the references.
Response: Thank Reviewer 2 very much for his/her comments. We have added the reference in right place (please see line 268 in page 8, and Ref 37).
“Ref [37] Li, Y. J.; Pu, H. T.; Wei, Y. L. Polypropylene/polyethylene multilayer separators with enhanced thermal stability for lithium-ion battery via multilayer coextrusion. Electrochim. Acta 2018, 264, 140-149.”
Round 2
Reviewer 1 Report
Comments and Suggestions for Authors
The major revision has been revised properly and It can be published